# U-DiTs: Downsample Tokens in U-Shaped Diffusion Transformers

**Yuchuan Tian**[1*], **Zhijun Tu**[2*], **Hanting Chen**[2], **Jie Hu**[2], **Chao Xu**[1], **Yunhe Wang**[2†]

[1] State Key Lab of General AI, School of Intelligence Science and Technology, Peking University.
[2] Huawei Noah's Ark Lab.

`tianyc@stu.pku.edu.cn`, `{zhijun.tu, chenhanting, hujie23, yunhe.wang}@huawei.com`
`xuchao@cis.pku.edu.cn`

## Abstract

Diffusion Transformers (DiTs) introduce the transformer architecture to diffusion tasks for latent-space image generation. With an isotropic architecture that chains a series of transformer blocks, DiTs demonstrate competitive performance and good scalability; but meanwhile, the abandonment of U-Net by DiTs and their following improvements is worth rethinking. To this end, we conduct a simple toy experiment by comparing a U-Net architectured DiT with an isotropic one. It turns out that the U-Net architecture only gain a slight advantage amid the U-Net inductive bias, indicating potential redundancies within the U-Net-style DiT. Inspired by the discovery that U-Net backbone features are low-frequency-dominated, we perform token downsampling on the query-key-value tuple for self-attention that bring further improvements despite a considerable amount of reduction in computation. Based on self-attention with downsampled tokens, we propose a series of U-shaped DiTs (U-DiTs) in the paper and conduct extensive experiments to demonstrate the extraordinary performance of U-DiT models. The proposed U-DiT could outperform DiT-XL/2 with only 1/6 of its computation cost. Codes are available at `https://github.com/YuchuanTian/U-DiT`.

## 1 Introduction

Thanks to the attention mechanism that establishes long-range spatial dependencies, Transformers [36] are proved highly effective on various vision tasks including image classification [15], object detection [5], segmentation [43], and image restoration [6]. DiTs [28] introduce full transformer backbones to diffusion, which demonstrate outstanding performance and scalability on image-space and latent-space generation tasks. Recent follow-up works have demonstrated the promising prospect of diffusion transformers by extending their applications to flexible-resolution image generation [26], realistic video generation [2], et cetera.

Interestingly, DiTs have discarded the U-Net architecture [30] that is universally applied in manifold previous works, either in pixel [20; 13] or latent space [29]. The use of isotropic (*i.e.* standard transformer; a plain stack of transformer blocks) architectures in DiTs is indeed successful, as scaled-up DiT models achieve supreme performance. However, the abandonment of the widely-applied U-Net architecture by DiTs and their improvements [18; 10; 26] on latent-space image generation tasks triggers our curiosity, because the U-Net inductive bias is always believed to help denoising. Hence, we rethink deploying DiTs on a canonical U-Net architecture.

In order to experiment with the combination of U-Net with DiT, we first propose a naive DiT in U-Net style (DiT-UNet) and compare it with an isotropic DiT of similar size. Results turn out that

---

[*]Equal Contribution.   [†]Corresponding Author.

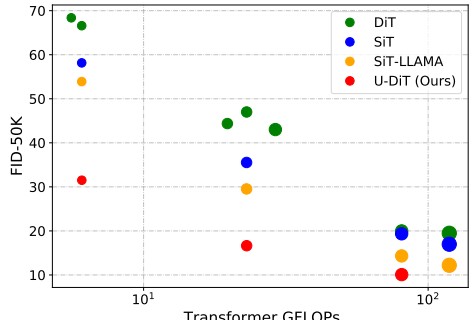 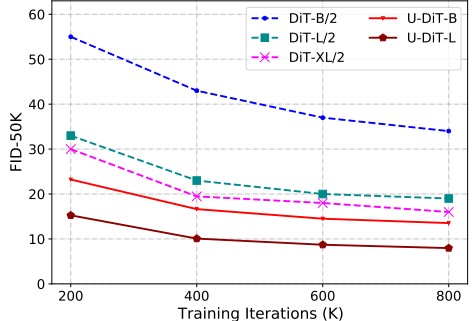

Figure 1: **Comparing U-DiTs with DiTs and their improvements.** We plot FID-50K versus denoiser GFLOPs (in log scale) after 400K training steps. U-DiTs could achieve better performance than its counterparts.

Figure 2: **The performance of U-DiTs and DiTs of various size.** U-DiTs perform consistently better than DiTs with the increase of training steps. The marker size represents the computation cost of the model qualitatively.

DiT-UNets are merely comparable to DiTs at similar computation costs. From this toy experiment, it is inferred that the inductive bias of U-Net is not fully leveraged when U-Nets and plain transformer blocks are simply combined.

Hence, we rethink the self-attention mechanism in DiT-UNet. The backbone in a latent U-Net denoiser provides a feature where low-frequency components dominate [31]. The discovery implies the existence of redundancies in backbone features: the attention module in the U-Net diffuser should highlight low-frequency domains. As previous theories praised downsampling for filtering high-frequency noises in diffusion [39], we seek to leverage this natural low-pass filter by performing token downsampling on the features for self-attention. Unlike previous transformer works [17; 44; 32] that downsample key-value pairs only, we radically downsample the query-key-value tuple altogether, such that self-attention is performed among downsampled latent tokens. It is surprising that when we incorporate self-attention with downsampled tokens into DiT-UNet, better results are achieved on latent U-Net diffusers with a significant reduction of computation.

Based on this discovery, we scale U-Nets with downsampled self-attention up and propose a series of State-of-the-Art U-shaped Diffusion Transformers (**U-DiT**s). We conduct manifold experiments to verify the outstanding performance and scalability of our U-DiT models over isotropic DiTs. As shown in Fig. 1 & Fig. 2, U-DiTs could outperform DiTs by large margins. Amazingly, the proposed U-DiT model could perform better than DiT-XL/2 which is 6 times larger in terms of FLOPs.

## 2 Preliminaries

**Vision Transformers.** ViTs [15] have introduced a transformer backbone to vision tasks by patchifying the input and viewing an image as a sequence of patch tokens and have proved its effectiveness on large-scale image classification tasks. While ViTs adopt an isotropic architecture, some following works on vision transformers [37; 25; 19; 40] adopt a pyramid-like hierarchical architecture that gradually downsamples the feature. The pyramid architecture is proved highly effective in classification and other downstream tasks. Apart from architectural improvements, some other works [3; 41] focuses on improving the Feed-Forward Network module in transformers.

Vision transformers are also mainstream backbones for denoising models. IPT [6] introduces an isotropic transformer backbone for denoising and other low-level tasks. Some later works [23; 22; 9] follow the isotropic convention, but other denoising works [38; 42] shift to U-Net backbones as their design. The pioneering work of U-ViT [1] and DiT [28] introduces full-transformer backbones to diffusion as denoisers.

**Recent Advancements in Diffusion Transformers.** Following DiTs, some works investigate the training and diffusion [16; 27] strategies of Diffusion Transformers. Other works focus on the design of the DiT backbone. DiffiT [8; 18] introduces a new fusion method for conditions; FiT [26] and VisionLLaMA [10] strengthens DiT by introducing LLM tricks including RoPE2D [34] and

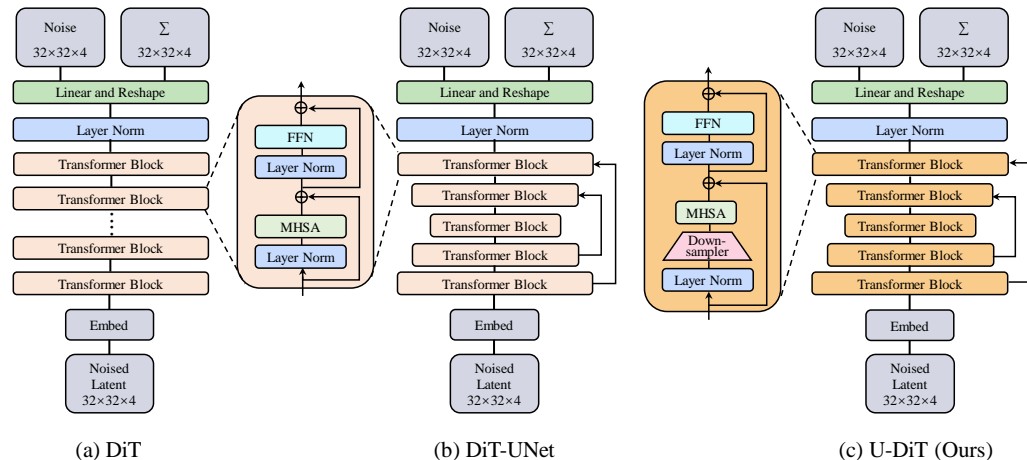

<p align="center">(a) DiT          (b) DiT-UNet          (c) U-DiT (Ours)</p>

Figure 3: **The evolution from the DiT to the proposed U-DiT.** Left (a): the original DiT, which uses an isotropic architecture. Middle (b): DiT-UNet, which is a plain U-Net-style DiT. We try this as a simple combination of DiT and U-Net in the toy experiment. Right (c): the proposed U-DiT. We propose to downsample the input features for self-attention. The downsampling operation could amazingly improve DiT-UNet with a huge cut on the amount of computation.

SwishGLU. These transformer-based diffusion works agree on adopting isotropic architectures on latents, *i.e.* the latent feature space is not downsampled throughout the whole diffusion model. The authors of DiT [28] even regard the inductive bias of U-Net as "not crucial".

**U-Nets for Diffusion.** From canonical works [20; 33; 13; 29], the design philosophy of U-Net [30] is generally accepted in diffusion. Specifically, Stable Diffusion [29] uses a U-Net-based denoiser on the compressed latent space for high-resolution image synthesis, which is highly successful in manifold generative tasks. Some previous trials on diffusion transformers [4; 18; 11; 21] also adopt U-Net on pixel-space generation tasks; but strangely, they shifted to isotropic DiT-like structures for latent-space diffusion. Despite its popularity in pixel-space diffusion, the U-Net architecture is not widely accepted in recent transformer-oriented works on latent-space diffusion.

Motivated by this, we are dedicated to investigating the potential of Transformer-backboned U-Net on latent-space diffusion. It is noteworthy that our goal is significantly different from U-ViT [1]: U-ViT is an isotropic transformer architecture with shortcuts, but our work resort to true U-Net architectures that involves multiple stages of feature-map downsampling and upsampling.

## 3 Investigating U-Net DiTs in Latent

As is recapped, the U-Net architecture is widely adopted in diffusion applications; theoretical evaluations on U-Net denoisers also reveal their advantage, as downsampling U-Net stage transitions could filter noises that dominate high frequencies [39]. The unprecedented desertion of isotropic architectures for latent diffusion transformers is thus counter-intuitive. We are rethinking and elucidating the potentials of transformer-backboned U-Net denoisers in latent diffusion via a toy experiment.

**A canonical U-Net-style DiT.** To start with, we propose a naive Transformer-backboned U-Net denoiser named **DiT-UNet** by embedding DiT blocks into a canonical U-Net architecture. Following previous U-Net designs, The DiT-UNet consists of an encoder and a decoder with an equal number of stages. When the encoder processes the input image by downsampling the image as stage-level amounts, the decoder scales up the encoded image from the most compressed stage to input size. At each encoder stage transition, spatial downsampling by the factor of 2 is performed while the feature dimension is doubled as well. Skip connections are provided at each stage transition. The skipped feature is concatenated and fused with the upsampled output from the previous decoder stage, replenishing information loss to decoders brought by feature downsampling. Considering the

small, cramped latent space ($32\times 32$ for $256\times256$-sized generation), we designate 3 stages in total, *i.e.* the feature is downsampled two times and subsequently recovered to its original size. In order to fit time and condition embeddings for various feature dimensions across multiscale stages, we use independent embedders for respective stages. In addition, we avoid patchifying the latent, as the U-Net architecture itself downsamples the latent space and there is no need for further spatial compression.

Via toy experiments, we compare the proposed U-Net-style DiT with the original DiT that adopts an isotropic architecture. In order to align the model with the DiT design, we repeatedly use plain DiT blocks in each stage. Each DiT block includes a self-attention module as the token mixer and a two-layer feed-forward network as the channel mixer. We conduct the experiment by training the U-Net-Style DiT for 400K iterations and compare it with DiT-S/4 which is comparable in size. All training hyperparameters are kept unchanged. It occurs that the U-Net style DiT only gains a limited advantage over the original isotropic DiT. The inductive bias of U-Net is insufficiently utilized.

| ImageNet 256×256 | | | | | | |
|---|---|---|---|---|---|---|
| Model | GFLOPs | FID↓ | sFID↓ | IS↑ | Precision↑ | Recall↑ |
| DiT-S/4 | 1.41 | 97.85 | 21.19 | 13.27 | 0.26 | 0.41 |
| DiT-UNet | 1.40 | 93.48 | **20.41** | 14.20 | 0.27 | 0.42 |
| DiT-UNet+Key-Value Downsampling | 0.91 | 94.38 | 23.21 | 14.32 | 0.27 | 0.40 |
| DiT-UNet+**Token Downsampling (Ours)** | **0.90** | **89.43** | 21.36 | **15.13** | **0.29** | **0.44** |

Table 1: **Toy experiments on U-Net-style DiTs.** The naive DiT-UNet performs slightly better than the isotropic DiT-S/4; but interestingly, when we apply token downsampling for self-attention, the DiT-UNet performs better with fewer costs.

**Improved U-Net-style DiT via token downsampling.** In seeking to incorporate attention in transformers to diffusion U-Nets better, we review the role of the U-Net backbone as the diffusion denoiser. A recent work on latent diffusion models [31] conducted frequency analysis on intermediate features from the U-Net backbone, and concluded that energy concentrates at the low-frequency domain. This frequency-domain discovery hints at potential redundancies in the backbone: the U-Net backbone should highlight the coarse object from a global perspective rather than the high-frequency details.

Naturally, we resort to attention with downsampled tokens. The operation of downsampling is a natural low-pass filter that discards high-frequency components. The low-pass feature of downsampling has been investigated under the diffusion scenario, which concludes that downsampling helps denoisers in diffusion as it automatically "discards those higher-frequency subspaces which are dominated by noise" [39]. Hence, we opt to downsample tokens for attention.

In fact, attention to downsampled tokens is not new. Previous works regarding vision transformers [17; 44] have proposed methods to downsample key-value pairs for computation cost reduction. Recent work on acceleration of diffusion models [32; 7] also applies key-value downsampling on Stable Diffusion models. But these works maintain the number of queries, and thus the downsampling operation is not completely performed. Besides, these downsampling measures usually involves a reduction of tensor size, which could result in a significant loss in information.

Different from these works, we propose a simple yet radical token downsampling method for DiT-UNets: we downsample queries, keys, and values at the same time for diffusion-friendly self-attention, but meanwhile we keep the overall tensor size to avoid information loss. The procedure is detailed as follows: the feature-map input is first converted into four $2\times$ downsampled features by the downsampler (the downsampler design is detailed in Sec. 4.2). Then, the downsampled features are mapped to $Q, K, V$ for self-attention. Self-attention is performed within each downsampled feature. After the attention operation, the downsampled tokens are spatially merged as a unity to recover the original number of tokens. Notably, the feature dimension is kept intact during the whole process. Unlike U-Net downsampling, we are not reducing or increasing the number of elements in the feature during the downsampling process. Rather, we send four downsampled tokens into self-attention in a parallel manner.

Self-attention with downsampled tokens does help DiT-UNets on the task of latent diffusion. As shown in Tab. 1, the substitution of downsampled self-attention to full-scale self-attention brings

slight improvement in the Fréchet Inception Distance (FID) metric despite a significant reduction in FLOPs.

**Complexity analysis.** Apart from the performance benefits, we are aware that adopting downsampled self-attention in the U-Shaped DiT could save as much as 1/3 of the model's overall computation cost. We conduct a brief computation complexity analysis on the self-attention mechanism to explain where the savings come from.

Given an input feature of size $N \times N$ and dimension $d$, we denote $Q, K, V \in \mathbb{R}^{N^2 \times d}$ as mapped query-key-value tuples. The complexity of self-attention is analyzed as:

$$X = \underbrace{AV}_{\mathcal{O}(N^4 D)} \qquad \text{s.t.} \qquad A = \textbf{Softmax}\underbrace{\left(QK^T\right)}_{\mathcal{O}(N^4 D)}.$$

In the proposed self-attention on downsampled tokens, four sets of downsampled query-key-value tuples $4 \times (Q_{\downarrow 2}, K_{\downarrow 2}, V_{\downarrow 2}) \in \mathbb{R}^{(\frac{N}{2})^2 \times d}$ performs self-attention respectively. While each self-attention operation only costs 1/16 of full-scale self-attention, the total cost for downsampled self-attention is 1/4 of full-scale self-attention. 3/4 of the self-attention computation is saved via token downsampling.

In a nutshell, we show from toy experiments that the redundancy of DiT-UNet is reduced by downsampling the tokens for self-attention.

## 4 Scaling the Model Up

Based on the discovery in our toy experiment, we propose a series of U-shaped DiTs (**U-DiT**) by applying the downsampled self-attention (proposed in Sec. 3) and scaling U-Net-Style DiT up.

**Settings.** We adopt the training setting of DiT. The same VAE (*i.e.* sd-vae-ft-ema) for latent diffusion models [29] and the AdamW optimizer is adopted. The training hyperparameters are kept unchanged, including global batch size 256, learning rate $1e - 4$, weight decay 0, and global seed 0. The training is conducted with the training set of ImageNet 2012 [12]. We used 8 NVIDIA A100s (80G) to train U-DiT-B and U-DiT-L models. The training overhead is listed in the appendix.

Apart from the self-attention on downsampling as introduced in the toy experiment (Section 3), we further introduce a series of modifications to U-DiTs, including cosine similarity attention [24; 22], RoPE2D [34; 26; 10], depthwise conv FFN [38; 3; 44], and re-parametrization [14; 35]. The contribution of each modification is quantitatively evaluated in Sec. 9.

### 4.1 U-DiT at Larger Scales

**Comparison with DiTs and their improvements.** In order to validate the effectiveness of the proposed U-DiT models beyond simple toy experiments, we scale them up and compare them with DiTs [28] of larger sizes. For a fair comparison, we use the same sets of training hyperparameters as DiT; all models are trained for 400K iterations. The results on ImageNet 256×256 are shown in Tab. 2, where we scale U-DiTs to $\sim 6e9$, $\sim 20e9$, $\sim 80e9$ FLOPs respectively and compare them with DiTs of similar computation costs, more details about the U-DiT architectures are shown in Tab. 8.

It could be concluded from Tab. 2 that all U-DiT models could outcompete their isotropic counterparts by considerable margins. Specifically, U-DiT-S and U-DiT-B could outperform DiTs of comparable size by $\sim 30$ FIDs; U-DiT-L could outperform DiT-XL/2 by $\sim 10$ FIDs. It is shocking that U-DiT-B could outcompete DiT-XL/2 with only 1/6 of the computation costs. In Tab. 3, we further demonstrate the advantage of U-DiTs over several competitive diffusion transformers [1; 28; 8; 18]. To present the advantage of our method better, we also include the performance of U-DiTs in an FID-50K versus FLOPs plot (Fig. 1). Apart from DiTs and U-DiTs, we also include other state-of-the-art methods: SiT [27] that proposes an interpolant framework for DiTs, and SiT-LLaMA [10] that combines state-of-the-art DiT backbone VisionLLaMA and SiT. The advantages of U-DiTs over other baselines are prominent in the plot. The results highlight the extraordinary scalability of the proposed U-DiT models.

**ImageNet 256×256**

| Model | FLOPs(G) | FID↓ | sFID↓ | IS↑ | Precision↑ | Recall↑ |
|---|---|---|---|---|---|---|
| **DiT-S/2** [28] | 6.06 | 68.40 | - | - | - | - |
| **DiT-S/2**[*] | 6.07 | 67.40 | 11.93 | 20.44 | 0.368 | 0.559 |
| **U-DiT-S (Ours)** | 6.04 | **31.51** | **8.97** | **51.62** | **0.543** | **0.633** |
| **DiT-L/4** [28] | 19.70 | 45.64 | - | - | - | - |
| **DiT-L/4**[*] | 19.70 | 46.10 | 9.17 | 31.05 | 0.472 | 0.612 |
| **DiT-B/2** [28] | 23.01 | 43.47 | - | - | - | - |
| **DiT-B/2**[*] | 23.02 | 42.84 | 8.24 | 33.66 | 0.491 | 0.629 |
| **U-DiT-B (Ours)** | 22.22 | **16.64** | **6.33** | **85.15** | **0.642** | **0.639** |
| **DiT-L/2** [28] | 80.71 | 23.33 | - | - | - | - |
| **DiT-L/2**[*] | 80.75 | 23.27 | 6.35 | 59.63 | 0.611 | **0.635** |
| **DiT-XL/2** [28] | 118.64 | 19.47 | - | - | - | - |
| **DiT-XL/2**[*] | 118.68 | 20.05 | 6.25 | 66.74 | 0.632 | 0.629 |
| **U-DiT-L (Ours)** | 85.00 | **10.08** | **5.21** | **112.44** | **0.702** | 0.631 |

Table 2: **Comparing U-DiTs against DiTs on ImageNet 256×256 generation.** Experiments with supermarks * are replicated according to the official code of DiT. We compare models trained for 400K iterations with the standard training hyperparameters of DiT. The performance of U-DiTs is outstanding: U-DiT-B could beat DiT-XL/2 with only **1/6** of inference FLOPs; U-DiT-L could outcompete DiT-XL/2 by 10 FIDs.

**ImageNet 256×256**

| Model | FLOPs (G) | FID↓ | sFID↓ | IS↑ | Precision↑ | Recall↑ |
|---|---|---|---|---|---|---|
| **U-ViT-L** [1] | 76.4 | 21.22 | 6.10 | 67.64 | 0.615 | 0.633 |
| **U-ViT-XL**[*] [1] | 113.0 | 18.35 | 5.75 | 76.59 | 0.632 | 0.630 |
| **DiT-XL/2** [28] | 118.7 | 20.05 | 6.25 | 66.74 | 0.632 | 0.629 |
| **PixArt-α-XL/2**[*] [8] | 118.4 | 24.75 | 6.08 | 52.24 | 0.612 | 0.613 |
| **DiffiT-XL/2**[*] [18] | 118.5 | 36.86 | 6.53 | 35.39 | 0.540 | 0.613 |
| **U-DiT-B (Ours)** | 22.2 | 16.64 | 6.33 | 85.15 | 0.642 | **0.639** |
| **U-DiT-L (Ours)** | 85.0 | **10.08** | **5.21** | **112.44** | **0.702** | 0.631 |

Table 3: **Comparing U-DiTs against competitive diffusion architectures on ImageNet 256×256 generation.** Since different architectures use different training settings, we align them under the official 400K-iteration setting of DiT for a fair comparison. The proposed U-DiT series could outperform these models by large margins at fewer FLOPs. Experiments with supermarks * include necessary modifications of the original work (detailed in the appendix).

U-DiTs are also performant in generation scenarios with classifier-free guidance. In Tab. 4, we compare U-DiTs with DiTs at $cfg = 1.5$. For a fair comparison, we train U-DiTs and DiTs for 400K iterations under identical settings.

**Extended training steps.** We evacuate the potentials of U-DiTs by extending training steps to 1 Million. Fig. 2 further demonstrate that the advantage of U-DiTs is consistent at all training steps. As training steps gradually goes up to 1 Million, the performance of U-DiTs is improving (Tab. 5). We

**ImageNet 256×256**

| Model | Cfg-Scale | FLOPs(G) | FID↓ | sFID↓ | IS↑ | Precision↑ | Recall↑ |
|---|---|---|---|---|---|---|---|
| **DiT-L/2**[*] | 1.5 | 80.75 | 7.53 | 4.78 | 134.69 | 0.780 | **0.532** |
| **DiT-XL/2**[*] | 1.5 | 118.68 | 6.24 | 4.66 | 150.10 | 0.794 | 0.514 |
| **U-DiT-B** | 1.5 | 22.22 | 4.26 | 4.74 | 199.18 | 0.825 | 0.507 |
| **U-DiT-L** | 1.5 | 85.00 | **3.37** | **4.49** | **246.03** | **0.862** | 0.502 |

Table 4: **Generation performance with classifier-free guidance.** We measure the performance of U-DiTs and DiTs at 400K training steps with $cfg = 1.5$. Experiments with a supermark * are replicated according to the official code of DiT. U-DiTs are also performant on conditional generation.

U-DiT-B U-DiT-L

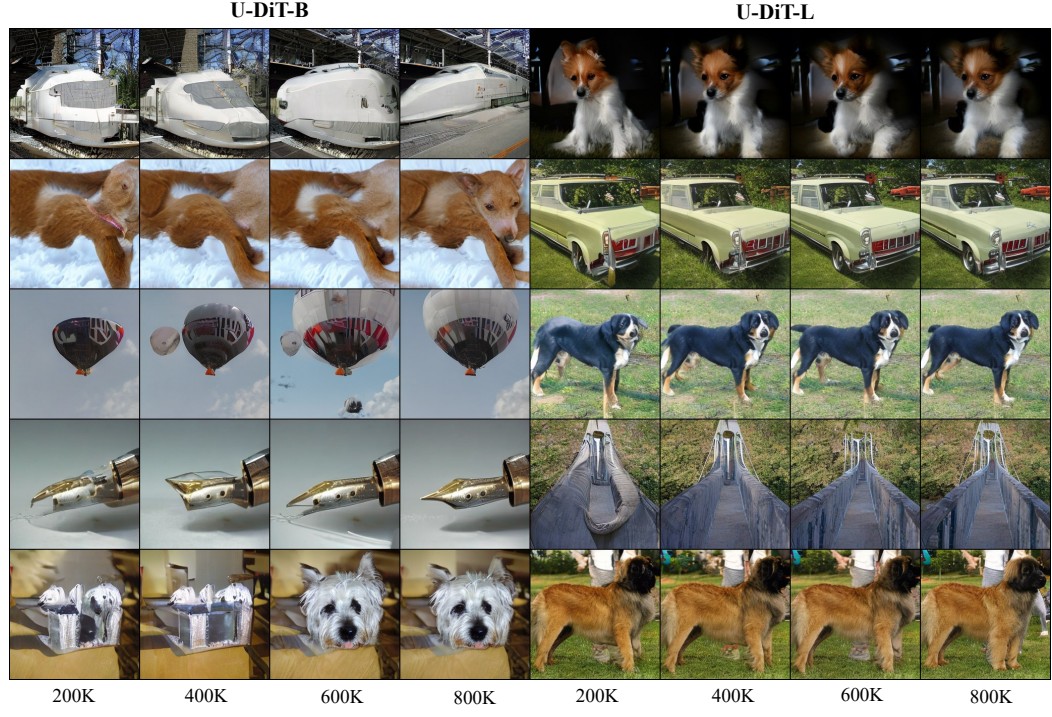

200K  400K  600K  800K  200K  400K  600K  800K

Figure 4: **Quality improvements of generated samples as training continues.** We sample from U-DiT models trained for different numbers of iterations on ImageNet 256×256. More training does improve generation quality. Best viewed on screen.

visualize the process where the image quality is gradually getting better (Fig. 4). Notably, U-DiT-L at only 600K training steps could outperform DiT-XL/2 at 7M training steps without classifier-free guidance. As additionally shown in Fig. 5, U-DiT models could conditionally generate authentic images at merely 1M iterations.

**Larger image size.** We additionally compare the generation performance of U-DiT-B and DiT-XL/2 on ImageNet $512 \times 512$ under exactly the same training setting. As shown in Tab. 6, U-DiT-B could still outcompete DiT-XL/2 that is approximately 5 times larger in FLOPs.

| ImageNet 256×256 | | | | | | |
|---|---|---|---|---|---|---|
| Model | Training Steps | FID↓ | sFID↓ | IS↑ | Precision↑ | Recall↑ |
| **DiT-XL/2** | 7M | 9.62 | - | - | - | - |
| **U-DiT-B** | 200K | 23.23 | 6.84 | 64.42 | 0.610 | 0.621 |
| **U-DiT-B** | 400K | 16.64 | 6.33 | 85.15 | 0.642 | 0.639 |
| **U-DiT-B** | 600K | 14.51 | 6.30 | 94.56 | 0.652 | 0.643 |
| **U-DiT-B** | 800K | 13.53 | **6.27** | 98.99 | 0.654 | 0.645 |
| **U-DiT-B** | 1M | **12.87** | 6.33 | **103.79** | **0.661** | **0.653** |
| **U-DiT-L** | 200K | 15.26 | 5.60 | 86.01 | 0.685 | 0.615 |
| **U-DiT-L** | 400K | 10.08 | 5.21 | 112.44 | 0.702 | 0.631 |
| **U-DiT-L** | 600K | 8.71 | **5.17** | 122.45 | 0.705 | 0.645 |
| **U-DiT-L** | 800K | 7.96 | 5.21 | 131.35 | 0.705 | 0.648 |
| **U-DiT-L** | 1M | **7.54** | 5.27 | **135.49** | **0.706** | **0.659** |

Table 5: **The performance of U-DiT-B and U-DiT-L models with respect to training iterations.** The unconditional generation performance of both models on ImageNet 256×256 consistently improves as training goes on, where U-DiT-L at 600K steps strikingly beats DiT-XL/2 at 7M steps.

| ImageNet 512×512 | | | | | | |
|---|---|---|---|---|---|---|
| Model | FLOPs (G) | FID↓ | sFID↓ | IS↑ | Precision↑ | Recall↑ |
| **DiT-XL/2**[*] | 524.7 | 20.94 | **6.78** | 66.30 | 0.745 | 0.581 |
| **U-DiT-B** | 106.7 | **15.39** | 6.86 | **92.73** | **0.756** | **0.605** |

Table 6: **Comparing U-DiTs against DiTs on ImageNet 512×512 generation.** Experiments with a supermark [*] are replicated according to the official code of DiT. We compare models trained for 400K iterations with the standard training hyperparameters of DiT.

| ImageNet 256×256 | | | | | | |
|---|---|---|---|---|---|---|
| Model | FLOPs(G) | FID↓ | sFID↓ | IS↑ | Precision↑ | Recall↑ |
| Pixel Shuffle (PS) | 0.89 | 96.15 | 23.90 | 13.93 | 0.272 | 0.389 |
| Depthwise (DW) Conv. + PS | 0.91 | 89.87 | **20.99** | 14.92 | 0.288 | 0.419 |
| **DW Conv. ‖ Shortcut + PS** | 0.91 | **89.43** | 21.36 | **15.13** | **0.291** | **0.436** |

Table 7: **Ablations on the choice of downsampler.** We have tried several downsampler designs, and it turns out that the parallel connection of a shortcut and a depthwise convolution is the best fit. We avoid using ordinary convolution (*i.e.* Conv.+PS) because channel-mixing is costly: conventional convolution-based downsamplers could double the amount of computation. The U-DiT with a conventional downsampler costs as many as 2.22G FLOPs in total.

## 4.2 Ablations

**The design of downsampler.** The downsampling operation in the proposed U-DiT transforms a complete feature into multiple spatially downsampled features. Based on previous wisdom, we figured out that previous works either directly perform pixel shuffling, or apply a convolution layer before pixel shuffling. While we hold that it is much too rigid to shuffle pixels directly as downsampling, applying convolution is hardly affordable in terms of computation costs. Specifically, ordinary convolutions are costly as extensive dense connections on the channel dimension are involved: using convolution-based downsamplers could double computation costs. As a compromise, we apply depthwise convolution instead. We also add a shortcut that short-circuits this depthwise convolution, which has proved crucial for better performance. The shortcut adds negligible computation cost to the model, and in fact, it could be removed during the inference stage with re-parameterization tricks. The results are shown in Tab. 7.

**The contribution of each individual modification.** In this part, we start from a plain U-Net-style DiT (DiT-UNet) and evaluate the contribution of individual components. Firstly, we inspect the advantage of downsampled self-attention. Recapping the toy experiment results in Sec. 3, replacing the full-scale self-attention with downsampled self-attention would result in an improvement in FID and 1/3 reduction in FLOPs. In order to evaluate the improvement of downsampling via model performance, we also design a slim version of DiT-UNet (*i.e.* DiT-UNet (Slim)). The DiT-UNet (Slim) serves as a full-scale self-attention baseline that spends approximately the same amount ($\sim$ 0.9GFLOPs) of computation as our U-DiT. As shown in the upper part of Tab. 9, by comparing U-DiT against DiT-UNet (Slim), it turns out that downsampling tokens in DiT-UNet could bring a performance improvement of $\sim$ 18FIDs.

Next, we inspect other modifications that further refine U-DiTs (lower part of Tab. 9). Swin Transformer V2 [24] proposes a stronger variant of self-attention: instead of directly multiplying Q and K matrices, cosine similarities between queries and keys are used. We apply the design to our self-attention, which yields $\sim$ 2.5FIDs of improvement. RoPE [34] is a powerful positional embedding method, which has been widely applied in Large Language Models. Following the latest diffusion transformer works [26; 10], we inject 2-dimensional RoPE (RoPE2D) into queries and keys right before self-attention. The introduction of RoPE2D improves performance by $\sim$ 2.5FIDs. Some recent transformer works strengthen MLP by inserting a depthwise convolution layer between two linear mappings [38; 3; 44]. As the measure is proved effective in these works, we borrow it to our U-DiT model, improving $\sim$ 5FIDs. As re-parametrization during training [14] could improve model performance, we apply the trick to FFN [35] and bring an additional improvement of $\sim$ 3.5FIDs. Above all, based on these components, the proposed U-DiTs are further improved.

Apart from the modifications that improve U-DiT, it is worth noting that vanilla U-DiTs (*i.e.* U-DiTs without any of the modifications mentioned above) are still competitive. According to Tab. 10, vanilla U-DiT-L could still achieve $\sim$ 8FIDs of advantage over DiT-XL/2.

| Model | Params (M) | FLOPs (G) | Channel | Head Number | Encoder-Decoder |
|---|---|---|---|---|---|
| **U-DiT-S** | 52.05 | 6.04 | 96 | 4 | [2, 5, 8, 5, 2] |
| **U-DiT-B** | 204.43 | 22.22 | 192 | 8 | [2, 5, 8, 5, 2] |
| **U-DiT-L** | 810.19 | 85.00 | 384 | 16 | [2, 5, 8, 5, 2] |

Table 8: **Configurations of U-DiTs architecture with different model sizes.** Channel represents the initial output channel number of first layer. Encoder-Decoder denotes the transformer block number of encoder and decoder module.

| ImageNet 256×256 | | | | | | |
|---|---|---|---|---|---|---|
| Model | FLOPs(G) | FID↓ | sFID↓ | IS↑ | Precision↑ | Recall↑ |
| **DiT-UNet** (Slim) | 0.92 | 107.00 | 24.66 | 11.95 | 0.230 | 0.315 |
| **DiT-UNet** | 1.40 | 93.48 | 20.41 | 14.20 | 0.274 | 0.415 |
| **U-DiT-T** (DiT-UNet+Downsampling) | **0.91** | 89.43 | 21.36 | 15.13 | 0.291 | 0.436 |
| **U-DiT-T** (+Cos.Sim.) | 0.91 | 86.96 | 19.98 | 15.63 | 0.299 | 0.450 |
| **U-DiT-T** (+RoPE2D) | 0.91 | 84.64 | 19.38 | 16.19 | 0.306 | 0.454 |
| **U-DiT-T** (+DWconv FFN) | 0.95 | 79.30 | 17.84 | 17.48 | 0.326 | 0.494 |
| **U-DiT-T** (+Re-param.) | 0.95 | **75.71** | **16.27** | **18.59** | **0.336** | **0.512** |

Table 9: **Ablations on U-DiT components.** Apart from the toy example in Sec. 3, we further validate the effectiveness of downsampled by comparing the U-DiT with a slimmed version of DiT-UNet at equal FLOPs. Results reveal that downsampling could bring $\sim$ 18FIDs on DiT-UNet. Further modifications on top of the U-DiT architecture could improve 2 to 5 FIDs each.

| ImageNet 256×256 | | | | | | |
|---|---|---|---|---|---|---|
| Model | FLOPs(G) | FID↓ | sFID↓ | IS↑ | Precision↑ | Recall↑ |
| **U-DiT-S** (Vanilla) | 5.91 | 41.01 | 10.96 | 39.29 | 0.489 | 0.622 |
| **U-DiT-S** (+All Mods) | 6.04 | **31.51** | **8.97** | **51.62** | **0.543** | **0.633** |
| **U-DiT-B** (Vanilla) | 21.96 | 20.89 | 7.33 | 72.85 | 0.611 | 0.637 |
| **U-DiT-B** (+All Mods) | 22.22 | **16.64** | **6.33** | **85.15** | **0.642** | **0.639** |
| **U-DiT-L** (Vanilla) | 84.48 | 12.04 | 5.37 | 102.63 | 0.684 | 0.628 |
| **U-DiT-L** (+All Mods) | 85.00 | **10.08** | **5.21** | **112.44** | **0.702** | **0.631** |

Table 10: **Comparison between vanilla U-DiTs and improved U-DiTs with all modifications.** With negligible extra computational overhead, the proposed modifications could improve the performance of U-DiT; but it is worth noting that vanilla U-DiTs are powerful enough against DiTs.

# 5 Conclusion

In this paper, we lay emphasis on DiTs in U-Net architecture for latent-space generation. Though isotropic-architectured DiTs have proved their strong scalability and outstanding performance, the effectiveness of the U-Net inductive bias is neglected. Thus, we rethink DiTs in the U-Net style. We first conduct an investigation on plain DiT-UNet, which is a straightforward combination of U-Net and DiT blocks, and try to reduce computation redundancy in the U-Net backbone. Inspired by previous wisdom on diffusion, we propose to downsample the visual tokens for self-attention and yield extraordinary results: the performance is further improved despite a huge cut on FLOPs. From this interesting discovery, we scale the U-Net architecture up and propose a series of U-shaped DiT models (U-DiTs). We have done various experiments to demonstrate the outstanding performance and scalability of our U-DiTs.

**Limitations.** For lack of computation resources and tight schedule, at this time we could not further extend training iterations and scale the model size up to fully investigate the potential of U-DiTs.

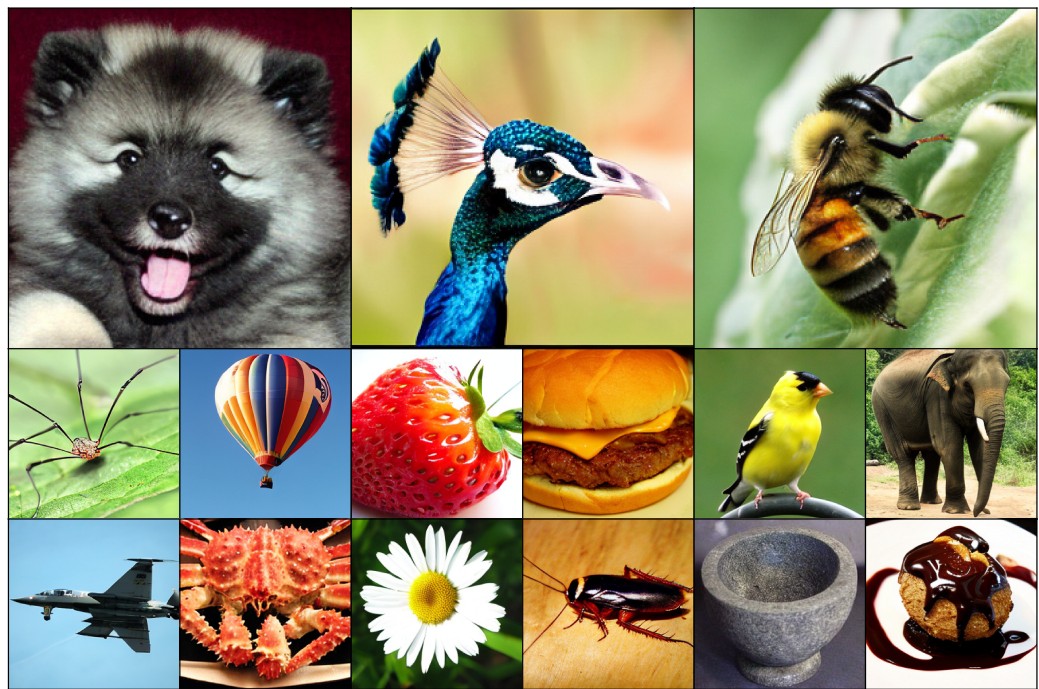

Figure 5: **Generated samples by U-DiT-L at 1M iterations.** It is astonishing that U-DiT could achieve authentic visual quality at merely 1 Million training steps. Best viewed on screen.

**Broader Impacts.** Due to the biases in the training data set, the generated content may contain pornographic, racist, hate and violent information. But we emphasize that the potential for misuse is mitigated through vigilant application.

**Discussion of Safeguards.** For cautionous usage, we suggest an algorithm capable of checking generated images, in order to identify and mitigate content that contravenes legal or ethical usages.

**Acknowledgement.** This work is supported by the National Key R&D Program of China under grant No. 2022ZD0160300 and the National Natural Science Foundation of China under grant No. 62276007. We gratefully acknowledge the support of MindSpore, CANN, and Ascend AI Processor used for this research.

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

# A  Appendix / supplemental material

## A.1  Details about Downsampling

Given an input tuple of (queries, keys, values) $QKV$ (shape=$(b, 3c, h, w)$), we firstly conduct Pixel-UnShuffle operation on $QKV$, and get four spatially downsampled $QKV$ (shape=$4 \times (bs^2, 3c, h/s, w/s)$). Then we perform vanilla multi-head self-attention, and get four downsampled output (shape=$4 \times (b \times s^2, c, h/s, w/s)$). Finally, we merge the four downsampled outputs into unity via Pixel-Shuffling (shape=$(b, c, h/s, w/s)$). Throughout the process, we not only significantly reduced the computational overhead of self-attention, but also ensured that the entire upsampling and downsampling process was completely lossless: the feature maps have not gone through lossy downsampling like bicubic or bilinear downsampling.

## A.2  Additional Experiment Details

**Training Overhead.** We report the training speed in Table 11. The training speed of vanilla U-DiT-L is comparable to that of DiT-XL/2.

| ImageNet $256 \times 256$ | | | | | | |
|---|---|---|---|---|---|---|
| Model | TS (Steps/Sec) | FID↓ | sFID↓ | IS↑ | Precision↑ | Recall↑ |
| **DiT-XL/2**[*] [28] | 1.71 | 20.05 | 6.25 | 66.74 | 0.632 | 0.629 |
| **U-DiT-B** (Vanilla) | 3.14 | 20.89 | 7.33 | 72.85 | 0.611 | 0.637 |
| **U-DiT-L** (Vanilla) | 1.55 | 12.04 | 5.37 | 102.63 | 0.684 | 0.628 |
| **U-DiT-L** (+All Mods) | 0.84 | **10.08** | **5.21** | **112.44** | **0.702** | **0.631** |

Table 11: **The training overhead of DiT-XL/2 and U-DiTs.** "TS" stands for training speed, measured in steps per second on 8 NVIDIA A100 (80G).

**Experiment Details in Table 3.** Since different diffusion architectures use different settings, we are dedicated to comparing them under identical settings for fair comparison. We adopt the 400K-iteration training setting of DiT-XL/2 [28]. Here are some further details regarding certain baselines:

1. **U-ViT-XL**: We increase the depth of U-ViT-L from 20 to 30 in order to match the FLOPs of DiT-XL/2. We encounter loss explosion while training U-ViT-H (133.25 GFLOPs) on the codebase of DiTs.

2. **PixArt-$\alpha$-XL/2**: As the original model is a text-to-image model, we removed its cross attention module for texts.

3. **DiffiT-XL/2**: This model is not open-sourced at the moment of this publication. Since it is a variant of DiT-XL/2, we replicated the time-dependent self-attention (TMSA) based on the codes of DiT. Unfortunately, the performance gets worse compared to the original DiT-XL/2.

**Additional Visual Results.** Due to large file size, we are unable to provide all visual results in the appendix. Please refer to the supplementary materials for two high-quality visual result demos.

