# OpenReview forum: "U-DiTs: Downsample Tokens in U-Shaped Diffusion Transformers"
_NeurIPS.cc/2024/Conference — NeurIPS 2024 poster_

### Official Review · Reviewer_JEXU · 2024-07-11

**Soundness:** 3
**Presentation:** 3
**Contribution:** 3
**Rating:** 6
**Confidence:** 5

**Summary:**

The paper presents a novel approach to enhancing diffusion models for image generation using Transformers. It introduces U-shaped Diffusion Transformers (U-DiTs) that employ token downsampling within the self-attention mechanism of U-Net architectures, aiming to reduce computational cost while maintaining or improving image generation quality. Computation efficiency is eagerly needed for the visual foundation model. This paper demonstrates that U-DiTs outperform traditional isotropic diffusion transformer models with significantly lower computation requirements. The paper also provides detailed ablations and analyses.

**Strengths:**

(1) The introduction of token downsampling in the self-attention process of U-Net style transformers is novel and addresses key efficiency issues in image generation tasks.

(2) The paper provides extensive experimental results showing U-DiTs achieving better performance metrics than larger, more computationally intensive models.

(3) The reduced computational demand of U-DiTs could make high-quality image generation more accessible and cost-effective.

**Weaknesses:**

(1) This paper needs to include important baselines. It fails to compare with other efficient DiT models, such as Pixart-sigma or others. SiT seems to be the only baseline method that takes flow as the objective. However, it does not employ a different architecture other than DiT, which should not be a very fair baseline to compare with U-DiTs.

(2) This paper only used ImageNet as the training data. No large-scale experiments have been conducted on Laion, JourneyDB, or others to verify its capacity for text-to-image generation. This limits the potential impact of this work.

(3) The operation of token downsampling is not clear. I am very confused about the "After the attention operation, the downsampled tokens are spatially merged as a unity to recover the original number of tokens." Fig 3 is also not clear since the down-sampler seems to be a black box. This is one of the most crucial parts of this paper. I would highly recommend the authors add more details on token merging, downsampling, and upsampling.

**Questions:**

(1) How do you get the GFLOPs as illustrated in the paper? How to estimate the GPU hours according to such GFLOPs?

---

> ### Author Rebuttal · Authors · 2024-08-07
>
> Dear reviewer JEXU,
>
> Thank you very much for your review. Here are our responses:
>
> **W1: Fail to compare with other efficient DiT models, like PixArt-Alpha.**
>
> Thanks for your advice. Here we provide a comparison with powerful baselines: PixArt-Alpha [1], as well as U-ViT [2] and DiffiT-XL [3]. We align all methods under the same setting as the standard setting (Table 2). The FLOPs statistics and metric results are shown in the Table below. Our method has a clear advantage over other methods.
>
> |                | GFLOPs   | FID       | sFID     | IS         | Precision | Recall    |
> | -------------- | -------- | --------- | -------- | ---------- | --------- | --------- |
> | DiffiT-XL      | 118.5    | 36.86     | 6.53     | 35.39      | 0.540     | 0.613     |
> | PixArt-Alpha   | 118.4    | 24.75     | 6.08     | 52.24      | 0.612     | 0.613     |
> | U-ViT-Large    | 76.4     | 21.22     | 6.10     | 67.64      | 0.615     | 0.633     |
> | U-DiT-B (Ours) | **22.2** | 16.64     | 6.33     | 85.15      | 0.642     | **0.639** |
> | U-DiT-L (Ours) | 85.0     | **10.08** | **5.21** | **112.44** | **0.702** | 0.631     |
>
> **W2: No large-scale experiments for text-to-image generation.**
>
> We are sorry that text-to-image generation training are too resource-intensive for us to conduct. We carefully consider this as a future work in order to verify its potentials on text-to-image generation.
>
> **W3: The operation of token downsampling is not clear. Please add more details on token merging, downsampling, and upsampling.**
>
> Thank you for your feedback. We appreciate your keen observations and will ensure to include these additional details in our revised manuscript. Actually, given a input tensor QKV_0 (shape=(b, 3c, h, w)), we firstly conduct PixelUnShuffle operation on QKV_0, and get four smaller feature QKV_1 (shape=(b$\times$s^2, 3c, h/s, w/s)). Then we perform vanilla multi-head self-attention, and get the output Y (shape=(b$\times$s^2, c, h/s, w/s)). Finally, we reshape the Y into the original shape (b, c, h, w). Throughout the process, we not only significantly reduced the computational overhead of self-attention, but also ensured that the entire upsampling and downsampling process was completely lossless. This is a key reason why our approach significantly outperforms the other methods listed in Table 1 of the paper.
>
> **Q1: How do you get the GFLOPs as illustrated in the paper? How to estimate the GPU hours according to such GFLOPs?**
>
> We got the GFLOPs via the Python library "torchprofile''[1], which is a "a general and accurate MACs / FLOPs profiler for PyTorch models". We are willing to provide training speed statistics in the table below. For fair comparison, we have removed all tricks to demonstrate the actual training time difference caused by the proposed architecture.
>
> |                   | GPU hours (400K) | FID       | sFID     | IS         | Precision | Recall    |
> | ----------------- | ---------------- | --------- | -------- | ---------- | --------- | --------- |
> | DiT-XL/2          | 519              | 20.05     | 6.25     | 66.74      | 0.632     | 0.629     |
> | U-DiT-L (Vanilla) | 573              | **12.04** | **5.37** | **102.63** | **0.684** | 0.628     |
> | U-DiT-B (Vanilla) | **283**          | 20.89     | 7.33     | 72.85      | 0.611     | **0.637** |
>
>
>
> Sincerely,
>
> Authors
>
> [1] "PixArt-α: Fast Training of Diffusion Transformer for Photorealistic Text-to-Image Synthesis." ICLR 2024.
>
> [2] "All are Worth Words: A ViT Backbone for Diffusion Models." CVPR 2023.
>
> [3] "DiffiT: Diffusion Vision Transformers for Image Generation." ECCV 2024.
>
> [4] https://github.com/zhijian-liu/torchprofile

---

> > ### Comment · Reviewer_JEXU · 2024-08-10
> > **Follow-up Question**
> >
> > Thank you for the detailed response. I have a follow-up question. What GPU that was used for your model training?

---

> ### Author Response · Authors · 2024-08-11
> **Authors' Response to Follow-up Question**
>
> Thank you for your helpful suggestions, and sorry for not specifying the devices. We used 8 NVIDIA A100s for training. The GPU-hours statistics were all measured on NVIDIA A100. We promise to add the results of more competitive baselines, as well as the table of FLOPs/GPU-Hour statistics, to Section 4 in later revisions.

---

> > ### Comment · Reviewer_JEXU · 2024-08-12
> >
> > 80G A100 or 40G A100?

---

> > > ### Author Response · Authors · 2024-08-12
> > > **Response to JEXU**
> > >
> > > Sorry for not making the config clear. We use 80G A100. We will add this detail to the manuscript in the next revision.

---

### Official Review · Reviewer_GsTx · 2024-07-11

**Soundness:** 4
**Presentation:** 3
**Contribution:** 3
**Rating:** 7
**Confidence:** 4

**Summary:**

This paper introduces a U-shaped diffusion Transformer (U-DiT) model, inspired by the departure from U-Net in DiT. The authors aim to combine the strengths of U-Net and DiT to determine if the inductive bias of U-Net can enhance DiTs. Initially, a simple DiT-UNet model was developed, but it showed minimal improvement over DiTs. Consequently, the authors explored downsampling QKV values in self-attention, which not only enhanced the model's performance but also significantly reduced computational costs. Leveraging this downsampled self-attention mechanism, a series of U-DiT models were proposed, with experimental results demonstrating their effectiveness.

**Strengths:**

The paper introduces an interesting idea that downsampled self-attention can reduce redundancy in U-Net while achieving performance improvements rather than losses. The authors support their claims with extensive experiments, demonstrating the convincing performance advantages of U-DiTs.

**Weaknesses:**

1. The statement in line 68, "the latent feature space is not downsampled," is incorrect for DiTs. DiTs reduce the spatial size of features by patchifying the latent features.
2. The paper presents several techniques that are claimed to be effective; however, their benefits decrease as the model scales up. For instance, in Table 8, the combined use of four techniques only results in a 2FID improvement for U-DiT-L.
3. While parameter count is a critical measure of model size, the paper does not provide a comparison of parameter counts with baseline models.

**Questions:**

1. Why do the tricks fail when the model is scaled up?
2. Please provide a comparison of parameter counts.

---

> ### Author Rebuttal · Authors · 2024-08-06
>
> Dear reviewer GsTx,
>
> Thank you very much for your comments. Here are our responses:
>
> **W1:** Statement "the latent feature space is not downsampled" is misleading.
>
> **A1**: Thanks for your suggestions. We will add qualifiers to this statement and limit it to the intermediate features.
>
> **W2&Q1:** Why do the tricks fail when the model scaled up?
>
> **A2:** Thank you for your insightful comments. (1) According to OpenAI’s scaling laws, model performance tends to be log-linear as model size increases. This means that as models become larger, the performance gains from the proposed tricks in our study, may appear less pronounced in absolute terms. This phenomenon occurs because larger models quickly approach the performance ceiling for the given evaluation metrics. (2) However, it is important to note that the improvements introduced by our tricks remain significant even for the largest models. These models are already nearing the upper limits of performance, and the relative improvements achieved by our methods are still impactful. Thus, while the absolute gain may be reduced, the effectiveness of our tricks is evident and noteworthy.
>
> **W3&Q2:** Please provide a comparison of parameter counts.
>
> **A3:** Thank you for your suggestion. The comparison of parameters is shown in the following table. With less FLOPs and a little more parameters, our proposed U-DiT-L still supress DiT-XL/2 significantly.
>
> | Model     | FLOPs (G)| Params (M) | FID |
> |:------------:|:----------------:|:--------:|:----------------:|
> | **U-DiT-B (Ours)** | **22.22** | **204.42** |  **16.64** |
> | DiT-L/2   | 80.73 | 458.10 |  23.33 |
> | DiT-XL/2   | 118.66  | 675.13  | 19.47 |
> | **U-DiT-L (Ours)** | **85.00** | **810.19** |  **10.08** |
>
> Typically, U-shaped models tend to have a larger number of parameters but lower computational overhead, whereas isotropic architecture models generally have higher computational requirements but fewer parameters. It is quite challenging to simultaneously align the parameter and computational overhead between these two types of models, and addressing this will be a primary focus of our future work.
>
> Sincerely,
>
> Authors

---

> > ### Comment · Reviewer_GsTx · 2024-08-11
> >
> > Thank you for the response and the additional experiments. In my view, the idea of combining U-Net with token downsampling for generation is very promising, as simple downsampling alone may not be sufficient for generative tasks. Besides, the FID@400K results look good. The authors also show the efficiency compared to other baselines.
> >
> > After considering the comments from other reviewers, I decided to raise my score to 7 (Accept).

---

> > > ### Author Response · Authors · 2024-08-11
> > > **Thanks**
> > >
> > > Thank you very much for your encouraging feedback, and thank you very much for your help in improving our paper as well!

---

### Official Review · Reviewer_AGQh · 2024-07-15

**Soundness:** 2
**Presentation:** 2
**Contribution:** 3
**Rating:** 4
**Confidence:** 4

**Summary:**

This paper proposes a transformer architecture as backbone for
diffusion modeling that is based on the UNet. The paper shows that a
variation of the transformer with downsampling layers and skip
connections achieves better results than the DiT at a lower
compoutational cost.

**Strengths:**

- Experimental results at multiple model scales show the proposed
  architecture obtains better performance than the original DiT
  architecture, at the same or lower computational cost
- The experiments suggest token downsampling for attention shows
  promising results and could be a worthile avenue for improvement.

**Weaknesses:**

- The novelty and technical contribution is limited. The proposed
  method consists of a minor architectural modification to the U-ViT
  architecture.
- Experiments are limited to ImageNet, and the FID scores achieved are
  far from state-of-the-art (FID<2).

**Questions:**

- There is a repeated use of the word isotropic, seemingly to refer to
  the standard transformer. I wonder what is the justification for
  this characterization.

**Limitations:**

Yes

---

> ### Author Rebuttal · Authors · 2024-08-07
>
> Dear reviewer AGQh,
>
> Thank you very much for your comments.  Here are our responses:
>
> W1.. **Novelty is limited.**
>
> Our work is not an improvement of U-ViT. The architecture of our U-DiT model is **completely different from U-ViT**: we adopt a U-Net architecture, while U-ViT is an isotropic architecture with skipping shortcuts. The major architectural difference is that U-DiT (Ours) is a encoder-decoder model, where each part have several stages where downsampling or upsampling is used as stage transition; U-ViT does **not involve any downsampling or upsampling**. The feature size is not changed throughout the whole model.
>
> We have also demonstrated via experiments that our model is much better than theirs.
>
> |                           | GFLOPs | FID       | sFID      | IS        | Precision | Recall   |
> | ------------------------- | ------ | --------- | --------- | --------- | --------- | -------- |
> | U-ViT-Large | 76.4   | 21.218 | 6.100 | 67.644 | 0.615 | 0.633 |
> | U-DiT-B (Ours) | **22.2** | 16.64     | 6.33     | 85.15      | 0.642     | **0.639** |
> | U-DiT-L (Ours) | 85.0     | **10.08** | **5.21** | **112.44** | **0.702** | 0.631     |
>
> W2. **About limited dataset and not reaching FID<2.**
>
> The reason we use ImageNet is that latest latent-space conditional generation models, like DiT, SiT, DiT-LLaMA, DiffiT all use ImageNet as the only benchmark. There are two versions of the ImageNet dataset: ImageNet-256 and ImageNet-512. As experiments in the paper are performed on ImageNet-256, we further add the performance of our model on ImageNet-512 to prove its robustness:
>
> |                           | GFLOPs | FID       | sFID      | IS        | Precision | Recall   |
> | ------------------------- | ------ | --------- | --------- | --------- | --------- | -------- |
> | DiT-XL | 524.7 | 20.94 | **6.78** | 66.30 | 0.645     | 0.581  |
> | U-DiT-B (Ours) | **106.7** | **15.39** | 6.86 | **92.73** | **0.756** | **0.605** |
>
> We are unable to pursue SOTA of FID<2 because it requires extremely large computation costs that we could not afford. For instance, DiT-XL needs 7M iterations to reach an FID of 2. We have compared with a SOTA model DiffiT (that reaches FID 1.73) under exactly the same 400K iteration setting. This indicates that our model is among the SOTA models in latent image generation.
>
> |                           | GFLOPs | FID       | sFID      | IS        | Precision | Recall   |
> | ------------------------- | ------ | --------- | --------- | --------- | --------- | -------- |
> | DiffiT | 118.5   | 36.862 | 6.533 | 35.391 | 0.540 | 0.613 |
> | U-DiT-B (Ours) | **22.2** | 16.64     | 6.33     | 85.15      | 0.642     | **0.639** |
> | U-DiT-L (Ours) | 85.0     | **10.08** | **5.21** | **112.44** | **0.702** | 0.631     |
>
> Q1. **The meaning of term "isotropic":** Yes, it refers to standard transformer where a stack of transformer blocks are concatenated in series. No downsampling or upsampling of the tokens is required within the model.
>
>
> Sincerely,
>
> Authors

---

> > ### Comment · Reviewer_AGQh · 2024-08-12
> > **Thank you for your response, and my remaining concerns.**
> >
> > I thank the authors for their response to my comments and to the other
> > reviewers.
> >
> > I still have some concerns about the paper:
> >
> > - Table 7 shows the additional modifications  are providing an
> >   improvement of 15 FID points, and that the downsampling alone is
> >   actually not that significant (although such high FID scores provide a
> >   weak signal). This raises the question of how much of the
> >   improvement in Table 2 is actually due to the downsampling
> >   introduced in Section 3, which is the main focus of the paper.
> >
> > - I am not fully convinced that filtering out high-frequency noise is
> >   always an advantage for diffusion, as claimed by the authors. In
> >   many diffusion settings one seeks to predict the noise, in which
> >   case this could be harmful.
> >
> > - The writing and terminology could be improved to clearly state which
> >   elements of the original U-Net are being referred to. When the
> >   authors say "we adopt a U-Net architecture", it can be confusing
> >   since the original U-Net is fully convolutional. As another example,
> >   I remain doubtful about the use of the word "isotropic", which
> >   should mean something that "is the same in every direction". However
> >   in this case it doesn't seem any particular latent dimension is
> >   treated differently.
> >
> > - Note there is the UVit architecture introduced in [1] which also
> >   includes downsampling/upsampling layers and signifcanlty outperforms
> >   DiT-XL (which is not a strong ImageNet baseline).
> >
> >
> > While I maintain these concerns, after reading the other reviewers I
> > realize the practical value this paper can have to the community in
> > providing directions to improve the transformer architecture for diffusion, and therefore raise
> > my score by 1 point.
> >
> >
> > [1] Hoogeboom, E., Heek, J. and Salimans, T., 2023, July. simple diffusion: End-to-end diffusion for high resolution images. In International Conference on Machine Learning (pp. 13213-13232). PMLR.

---

> > > ### Author Response · Authors · 2024-08-13
> > > **Thank You and Further Responses**
> > >
> > > Thanks for your approval and thanks again for your helpful suggestions. Here are our responses:
> > >
> > > ### Q1: Limited improvement in Table 7.
> > >
> > > The key is that we also need to take model **FLOPs** into consideration in comparison. In Table 7, though the proposed downsampling could improve a mere 4 FID, it reduces the FLOPs by **more than 1/3** of the DiT-UNet model. In order to evaluate the advantage of downsampling through FID improvement, we evaluated a smaller "DiT-UNet (Slim)" model that is comparable in FLOPs to U-DiT-T. The advantage measured under comparable FLOPs is around **28 FID**, which is nearly **twice** the improvement of all the tricks. Below we provide a copy of Table 7 (upper section) for easy reference:
> > >
> > > |                                     | GFLOPs | FID       | sFID      | IS        | Precision | Recall    |
> > > | ----------------------------------- | ------ | --------- | --------- | --------- | --------- | --------- |
> > > | **DiT-UNet (Slim)**                 | 0.92   | 107.00    | 24.66     | 11.95     | 0.230     | 0.315     |
> > > | DiT-UNet                            | 1.40   | 93.48     | **20.41** | 14.20     | 0.274     | 0.415     |
> > > | **U-DiT-T (DiT-UNet+Downsampling)** | 0.91   | **89.43** | 21.36     | **15.13** | **0.291** | **0.436** |
> > >
> > > ### Q2: Filtering out high-frequency noise is harmful.
> > >
> > > We provide a further explanation as follows: estimating noise is equivalent to estimating clear image via simple transformation, which implies that perception of clear denoised signal is vital in the denoiser. Additionally, shortcuts passes most high-frequencies, while the backbone is low-frequency dominated [1]. Hence, filtering out high-frequencies in the backbone is not causing significant impacts.
> > >
> > > ### Q3: Terminology issues with "U-Net" and "Isotropic".
> > >
> > > Thank you for suggestions on terminology. We agree that the term "U-Net" may cause misunderstanding. After referring to Ronneberger et al. [2], we hold that "U-Shaped Architecture/Network" is a better substitute. However, we found it hard to find a term substitute to "Isotropic". We will explain it as "a standard transformer architecture that does not involve any change in token size" in the next revision.
> > >
> > > ### Q4: Comparison to UViT (in Simple Diffusion [3]).
> > >
> > > We hold that our method is different from UViT [3] as follows:
> > >
> > > 1. Task difference: UViT experiments are conducted on **pixel-space**; DiT and U-DiT experiments are conducted on **latent-space**.
> > >
> > > 2. Training setting difference: UViT is using **batch-size 2048** for 500K iterations on the ImageNet-256 benchmark; DiT and our U-DiT uses batch-size 256 (which is only **1/8** of the batch-size of UViT). The authors of [3] themselves claim that "the batch size is larger (2048) which does affect FID and IS performance considerably".
> > >
> > > 3. Model size difference: UViT has **2 Billion** parameters, which is more than **2 times** the size of the largest variant of U-DiT (U-DiT has only 810M).
> > >
> > > 4. Architectural difference: UViT **only** uses Transformer Block at the medium stage; it keeps using **conventional ResBlock** at the encoder-decoder stage (Fig. 7 in [3]). Our U-DiT model uses **Transformer Blocks across all stages**.
> > >
> > > Above all, we hold that UViT is **not fairly comparable** to DiT and U-DiT, but we do thank the reviewer for providing a competitive diffusion architecture, and we will discuss it in Section 2: Preliminaries in the next revision. Additionally, we sincerely apologize for not being able to test the performance UViT on the same setting of DiT due to limited time left for discussion and closed-source UViT codes.
> > >
> > > [1] Freeu: Free lunch in diffusion u-net. CVPR 2024.
> > >
> > > [2] U-Net: Convolutional Networks for Biomedical Image Segmentation. MICCAI 2015.
> > >
> > > [3] Simple diffusion: End-to-end diffusion for high resolution images. ICML 2023.

---

> > > > ### Author Response · Authors · 2024-08-14
> > > > **Thank you to reviewer AGQh**
> > > >
> > > > Dear reviewer AGQh,
> > > >
> > > > At the very end of the discussion period, we want to express our thanks to reviewer AGQh for their help in improving our paper. May I kindly ask if there is any other concerns that we could help you? Thank you!

---

### Official Review · Reviewer_Df8i · 2024-07-30

**Soundness:** 3
**Presentation:** 3
**Contribution:** 2
**Rating:** 5
**Confidence:** 4

**Summary:**

The authors conduct a simple toy experiment by comparing a U-Net architectured DiT with an isotropic one. They find that the U-Net architecture only gains a slight advantage, indicating potential redundancies within the U-Net-style DiT.  Inspired by the discovery that U-Net backbone features are low-frequency-dominated, they perform token downsampling on the query-key-value tuple for self-attention and observed performance improvements.

Based on self-attention with downsampled tokens, the authors propose a series of U-shaped DiTs (U-DiTs) in the paper and conduct extensive experiments to demonstrate the good performance of U-DiT models.

**Strengths:**

- The author verified that radical token downsampling method for DiT-UNets could save the overall computation cost compared to full-scale self-attention, while can still improve the performance.

- the authors also performed scaling experiments to compare with various scaled DiTs.

**Weaknesses:**

- there is a lack of comparison with some noticeable existing works, e.g., the authors did not compare with U-ViT, which is pretty similar as the plain UNet based DiT. There is also no mentioning or comparison with HourglassDiT.
- there is no comparison with reducing tokens by simply increasing the patch size.
- the saved computation is not clear, line 141 says 1/3 is saved while 149 claims 3/4.
- there is no justification or explanation why radically downsampling of K, V, Q is better than only k-v downsampling.
- there is no explanation on why radically reduced tokens would gain better performance.

**Questions:**

- if the downsampling factor is 2, how much layers will this downsampled design support?

**Limitations:**

The authors may not have to pursue even longer iterations and comparison with DiTs. More justification on why downsampling tokens would help is more important. Also extending the experiments other than ImageNet would be better.

---

> ### Author Rebuttal · Authors · 2024-08-07
>
> Dear reviewer Df8i,
>
> Thank you very much for your suggestions. Here are our responses:
>
> **W1.** We omitted U-ViT and Hourglass DiT previously, because these models are mainly targeted at pixel-space generation, while our work is focused on latent space generation. To demonstrate the advantage of our model, we conducted experiments as follows:
>
> |                           | GFLOPs | FID       | sFID      | IS        | Precision | Recall   |
> | ------------------------- | ------ | --------- | --------- | --------- | --------- | -------- |
> | DiffiT | 118.5   | 36.862 | 6.533 | 35.391 | 0.540 | 0.613 |
> | PixArt-XL | 118.4   |   24.751 | 6.075 | 52.237 | 0.612 | 0.613 |
> | U-ViT-Large | 76.4   | 21.218 | 6.100 | 67.644 | 0.615 | 0.633 |
> | Hourglass-DiT * | 53.8    | 564.574 | 747.528 | 1.000 | 0.000 | 0.000 |
> | U-DiT-B (Ours) | **22.2** | 16.64     | 6.33     | 85.15      | 0.642     | **0.639** |
> | U-DiT-L (Ours) | 85.0     | **10.08** | **5.21** | **112.44** | **0.702** | 0.631     |
>
> P.S. H-DiT is intended for pixel-space generation instead of latent space generation (which is the usage of our U-DiT model). The officially provided H-DiT models are very small (because they are intended for High-resolution pixel space generation), so we scale them up for 12x (which is the maximal size we could scale). The training of the enormously large H-DiT is not stable.
>
>
>   Plus, we want to stress that **U-ViT is not a U-Net architecture**: it is an isotropic architecture with shortcuts that does not involve feature downsampling. Our U-DiT model, on the other hand, is a U-Net architecture with feature downsampling. Besides feature downsampling, we are further adding attention downsampling to the self-attention module.
>
> **W2.** Thank you very much for your suggestions. Following the setting of Table 1, we have conducted an experiment that uses patchsize 2 as follows:
>
> |                           | GFLOPs | FID       | sFID      | IS        | Precision | Recall   |
> | ------------------------- | ------ | --------- | --------- | --------- | --------- | -------- |
> | Token Patchification      | 0.88   | 129.12    | 34.02     | 9.30      | 0.17      | 0.21     |
> | Token Downsampling (Ours) | 0.90   | **89.43** | **21.36** | **15.13** | **0.29**  | **0.44** |
>
> Results reveal that patchsize=2 performs worse than our proposed downsampling method.
>
> **W3.** We apologize for the ambiguities. "1/3" in line 141 refers to FLOPs saved in total (on the entire model); "3/4" in line 149 refers to FLOPs saved within self-attention. Apart from self-attention, there are many other components in the model. We will clarify the meaning of these fractions in the next revision.
>
> **W4.** We have provided a brief explanation in line 121-123, and here is some further clarification: KV Compression keeps the number of queries intact (which corresponse to the number of output tokens), which means downsampling is not performed completely on the feature map; the effect of noise filtering is thus reduced. We have downsampled queries for self-attention.  Besides, KV Downsampling involves a lossy reduction of Key-Value pair tokens; our Token Downsampling measure does not involve lossy reduction of tensors.
>
>
> **Q1.** All multi-head self-attention (MHSA) layers in U-DiT support downsampling tokens by 2, and we did apply downsampling in all MHSA layers in actual practice.
>
>
> Sincerely,
>
> Authors

---

> > ### Author Response · Authors · 2024-08-07
> > **Additional Comments on "why radically reduced tokens would gain better performance"**
> >
> > Sorry for not making why "reduced tokens would gain better performance" clear. On one hand, low-frequency dominates the backbone of U-Net, and thus downsampling would cause little information loss; on the other hand, downsampling could filter out high-frequency noises (according to line 115-117) and thus being beneficial for diffusion. That's why reduced tokens would gain better performance.

---

### Decision · Program_Chairs · 2024-09-25

**Decision:**

Accept (poster)

**Comment:**

This paper improves UNet based on downsampling layers and skip connections for diffusion modeling. The paper received diverging ratings with a common concern around limited novelty. It is indeed not the most original idea to add downsampling/upsampling and skip connections to a neural architecture. But the paper does show convincing results that prove token downsampling inside the transformer is better than patchification at the input for a DiT. Hence this paper could inform future efforts on improving the popular DiT architectures and training recipes. Other than that, the majority of reviewers found that their concerns were sufficiently addressed during discussion and recommend acceptance. The AC shares the majority opinion.

It is recommended to integrate all the discussions/comparisons to the final paper. Furthermore, it's key to address the remaining concern from reviewer AGQh on the performance and scope of experiments. Given that the performance is far from the SOTA frontier, the authors argue for the trade-off between FID and training cost/model size. Then the authors need to answer one important question in the final paper: would this approach allow achieving the highest sample quality, or would the proposed token downsampling limit the model capacity at some point?